# Unsupervised Online Grounding for Social Robots

**Oliver Roesler** [1,*] and **Elahe Bagheri** [2]

1    Artificial Intelligence Lab, Vrije Universiteit Brussel, 1050 Brussels, Belgium
2    Robotics and Multibody Mechanics Research Group, Vrije Universiteit Brussel and Flanders Make, 1050 Brussels, Belgium; elahe.bagheri@vub.be
\*    Correspondence: oliver@roesler.co.uk

**Abstract:** Robots that incorporate social norms in their behaviors are seen as more supportive, friendly, and understanding. Since it is impossible to manually specify the most appropriate behavior for all possible situations, robots need to be able to learn it through trial and error, by observing interactions between humans, or by utilizing theoretical knowledge available in natural language. In contrast to the former two approaches, the latter has not received much attention because understanding natural language is non-trivial and requires proper grounding mechanisms to link words to corresponding perceptual information. Previous grounding studies have mostly focused on grounding of concepts relevant to object manipulation, while grounding of more abstract concepts relevant to the learning of social norms has so far not been investigated. Therefore, this paper presents an unsupervised cross-situational learning based online grounding framework to ground emotion types, emotion intensities and genders. The proposed framework is evaluated through a simulated human–agent interaction scenario and compared to an existing unsupervised Bayesian grounding framework. The obtained results show that the proposed framework is able to ground words, including synonyms, through their corresponding perceptual features in an unsupervised and open-ended manner, while outperforming the baseline in terms of grounding accuracy, transparency, and deployability.

**Keywords:** language grounding; cross-situational learning; online learning

## 1. Introduction

The number of humans who interact with robots in a social context is increasing. In contrast to industrial robots, the purpose of social robots is not the accurate performance of a specific task in a highly constrained environment, but to assist in therapy, education or social services [1]. Nowadays, social robots are already employed in many different areas to help elderly people [2,3], people with dementia [4], autistic children [5] as well as people with disabilities [6]. Previous research showed that social robots that follow human social norms, like empathy, are seen as more supportive [7] and friendly [8]. A social norm defines how people should behave, thereby defining specific expectations, which, when violated, lead to specific reactions, including sanctions or punishment [9,10]. Thus, enabling robots to follow social norms has the potential to make human–robot interactions more enjoyable, predictable, natural, and, in general, more similar to interactions between humans. However, due to the large number of social norms, their dynamic nature, i.e., they can change over time, and their strong variation based on the environment of the interaction as well as the personality of the human the robot is interacting with, they cannot be hard-coded into the robot but must instead be learned. Learning can either occur through trial-and-error, i.e., reinforcement learning [11], by observing how humans interact with each other, i.e., learning from demonstration [12], or by utilizing abstract knowledge available in written form, e.g., on the web or in books. While there have been several studies that investigated learning from demonstration and reinforcement learning to learn social norms [13–16], there have not been any studies that investigated learning of social norms from abstract knowledge provided in natural language. Understanding natural language is non-trivial

and requires sophisticated language grounding mechanisms that provide meaning to language by linking words and phrases to corresponding concrete representations, which represent sets of invariant perceptual features obtained through an agent's sensors that are sufficient to distinguish percepts belonging to different concepts [17]. Most grounding research has focused on understanding natural language instructions so that robots can identify and manipulate the correct object [18,19] or navigate to the correct destination [20], while, to the best of our knowledge, no attempts have been made to ground more abstract concepts, such as emotion types, emotion intensities and genders, which are essential to understand natural language texts describing social norms, such as empathy.

In this study, we try to fill this gap by proposing an unsupervised online grounding framework, which uses cross-situational learning to ground words describing emotion types, emotion intensities and genders through their corresponding concrete representations extracted from audio with the help of deep learning. The proposed framework is evaluated through a simulated human–agent interaction experiment in which the agent listens to the speech of different people and receives at the same time a natural language description, describing the gender of the observed person as well as the experienced emotion. Furthermore, the proposed framework is compared to a Bayesian grounding framework that has been employed in several previous studies to ground words through a variety of different percepts [18–21].

The remainder of this paper is structured as follows: the next section provides some background regarding cross-situational learning. Afterwards, Section 3 discusses related work in the area of language grounding. The proposed framework, the baseline and the employed experimental setup are explained in Sections 4–6. Section 7 describes the obtained results. Finally, Section 8 concludes the paper.

## 2. Background

Cross-situational learning (CSL) has been proposed by, among others, Pinker [22] and Fisher et al. [23] as a mechanism to learn the meaning of words by tracking their co-occurrences with concrete representations of percepts across multiple situations, which enables it to handle referential uncertainty. The basic idea is that the context a word is used in leads to a set of candidate meanings, i.e., mappings from words to concrete representations, and that the correct meaning lies where the sets of candidate meanings intersect. Thus, the correct mapping between words and their corresponding concrete representations can be identified through repeated co-occurrences so that the learner is able to select the meaning which reliably reoccurs across situations [24,25]. The original idea of CSL was developed to explain the remarkable ability of human children to learn the meaning of words without any prior knowledge of natural language. Afterwards, a number of experimental studies [26–28] have confirmed that CSL is, in fact, used by humans to learn the meaning of words. Since CSL requires the learner to be exposed more than one time to a word to learn its meaning, it belongs to the group of slow-mapping mechanisms through which most words are acquired [29]. In contrast, fast-mapping mechanisms that enable the acquisition of word meanings through a single exposure are only used for a limited number of words [30,31]. The successful use of CSL by humans has inspired the development of many different CSL-based grounding algorithms (Section 3) to enable artificial agents to learn the meaning of words by grounding them through corresponding concrete representations obtained with their sensors.

## 3. Related Work

Since Harnad [17] proposed the "Symbol Grounding Problem", a variety of models which either utilize unsupervised or interactive learning mechanisms have been proposed to create connections between words and corresponding concrete representations. Interactive learning approaches are based on the assumption that another agent is available that already knows the correct connections between words and concrete representations so that it can support the learning agent by providing feedback and guidance. Due to

this support, interactive learning models are usually faster, and often also more accurate than unsupervised learning models; however, they do not work in the absence of a tutor who provides the required support. Furthermore, in most studies, e.g., [32,33], the tutoring agent did not provide real natural language sentences but only single words, which significantly simplifies the grounding problem and raises the question of whether these models would work outside the laboratory. Since, in real environments, the tutor would be a regular user and might not be aware of the limitations of the learning agent or unwilling to adjust the interaction accordingly. Examples of interactive grounding approaches include the "Naming Game" [34], which has been used in many studies to ground a variety of percepts, such as colors or prepositions [32,33], and the work by She et al. [35]. The latter used a dialog system to ground higher level symbols through already grounded lower level symbols, thereby introducing another constraint that a sufficiently large set of already grounded lower level symbols is available. It is important to note that most, if not all, existing interactive learning approaches assume that the provided support is always correct, although it might be wrong due to noise or malicious intent of the tutoring agent. In contrast to interactive-learning-based approaches, unsupervised grounding approaches do not require any form of supervision and learn the meaning of words across multiple exposures through cross-situational learning [36,37]. The main advantage is that no tutor is needed, which makes them more easy to deploy and also removes a potential source of noise because it cannot be guaranteed that another agent that is able and willing to act as a tutor is present, nor can it be assumed that the received support is always correct. Both points are important when deploying an agent in a dynamic uncontrolled environment that does not allow any control over the people who interact with the agent. In previous studies, cross-situational learning has been used for grounding of shapes, colors, actions, and spatial concepts [18,20,21]. However, most proposed models only work offline, i.e., perceptual data and words need to be collected in advance, and the employed scenarios only contained unambiguous words, i.e., no two words were grounded through the same percept. In contrast, the grounding framework used in this study, which is based on the framework proposed in [38], is able to learn online and in an open-ended manner, i.e., no separate training phase is required, and it is also able to ground synonyms, i.e., words that refer to the same concrete representations in specific context, e.g., "happy" and "cheerful".

## 4. Proposed Framework

The proposed framework consists of three parts: (1) Perceptual feature extraction component, which extracts audio features from video using openEAR [39], (2) Perceptual feature classification component, which uses deep neural networks to obtain concrete representations of perceptual features, (3) Language grounding component, which creates mappings from words to corresponding concrete representations using cross-situational learning. The individual parts of the employed framework are illustrated below and described in detail in the following subsections.

1. Perceptual feature extraction
   - **Input**: Video stream.
   - **Output**: 156 audio features.
2. Perceptual feature classification
   - **Input**: 156 audio features.
   - **Output**: Concrete representations of percepts.
3. Language grounding
   - **Input**: Natural language descriptions, concrete representations of percepts, previously detected auxiliary words, and word and percept occurrence information.
   - **Output**: Set of auxiliary words and word to concrete representation mappings.

### 4.1. Perceptual Featue Extraction

Perceptual features are extracted from the videos of The Ryerson Audio-Visual Database of Emotional Speech and Song (RAVDESS) [40] since it is used in this study to simulate human–robot interactions (Section 6). All videos are given directly, i.e., without any preprocessing, as input to openEAR [39], which is a freely available open-source toolkit, to extract 384 speech features including the minimum, maximum, and mean values for each individual speech feature. Which features are extracted by openEAR depends on the used configuration. Three different configurations, i.e., INTERSPEECH 2009, emobase and INTERSPEECH 2013, were evaluated for this study but only the INTERSPEECH 2009 (emo-IS09) [41] configuration was used in the end because its features led to the best classification results. The available feature sets are pulse code modulation (PCM) root mean square (RMS) frame energy, mel-frequency cepstral coefficients (MFCC), PCM zero-crossing rate (ZCR), voice probability (voiceProb), and F0. Additionally, for each of the mentioned feature sets, a corresponding set with the delta coefficients is provided [41]. However, only the MFCC and PCM RMS features, i.e., 156 of the 384 obtained features, are provided to the classification models (Section 4.2) because they produced the best classification results based on an experimental evaluation of the available feature sets, i.e., each feature set and different combinations of feature sets were provided to the employed models. For the evaluation, the mean accuracies calculated across five runs for each feature set combination were compared and the model of the best-performing run was used in this study to obtain the concrete representations provided as input to the language grounding component.

### 4.2. Perceptual Feature Classification

For the classification task, the dataset is partitioned in a subject independent manner into a train and test set, i.e., the videos of the first eighteen actors (nine female, nine male) are used for training and the videos of the remaining six subjects (three female, three male) are used for testing (Section 6). The 156 audio features extracted by openEAR (Section 4.1) are used as input for three different deep learning models, i.e., one for each modality, after being normalized betwen zero and one. For emotion type classification, the model consists of four dense layers each followed by a dropout layer with a ratio of 0.1. The batch size and epoch size are set to 160 and 250, respectively. ReLU is used as an activation function in the first three dense layers, a Softmax function is used in the last layer, and Adam is used as an optimizer [42]. The applied model obtained an accuracy of 59.6% when classifying six basic emotions and neutral.

For emotion intensity recognition and gender recognition, the model proposed in [43] is used (Figure 1) with the following parameter settings: the convolutional layers are all 1D, have a kernels of size 3 and use ReLU as activation functions to add non-linearity. The dropout layers are used as regularizers with a ratio of 0.1. The 1D max-pooling layers have a kernel size of four and are used to introduce sparsity in the network parameters and to learn deep feature representations. Finally, the dense layers are used with sigmoid activation functions to find the predicted binary distribution of the target class. The number of epochs is 250 and the batch size is set to 128. The number of units in applied LSTM and BiLSTM networks is five. The applied model obtained an accuracy of 89.8% for gender recognition and 73.5% for emotion intensity recognition. Table 1 provides an overview of the classification accuracies for all individual classes.

**Table 1.** Classification accuracies for all individual classes, i.e., concrete representations.

| Emotion Type | | | | | | | Emotion Intensity | | Gender | |
|---|---|---|---|---|---|---|---|---|---|---|
| **Happiness** | **Sadness** | **Anger** | **Neutral** | **Surprise** | **Fear** | **Disgust** | **Normal** | **Strong** | **Male** | **Female** |
| 50% | 77.08% | 77.08% | 37.5% | 62.5% | 52.08% | 56.25% | 41.66% | 80.5% | 99.35% | 78.02% |

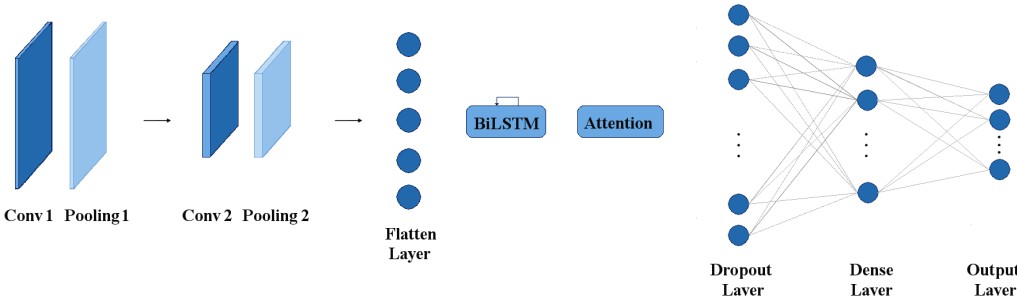

**Figure 1.** The architecture of the classification model used for emotion intensity and gender recognition.

### 4.3. Language Grounding

The grounding algorithm described in this section requires percepts to be represented through concrete representations. In previous work, concrete representations have been obtained through clustering [38]; however, the clustering algorithms employed in previous studies are not able to achieve accurate clusters for the extracted speech features. Thus, deep neural networks are used instead to obtain the same label for all concrete representations of the same emotion type, emotion intensity or gender (Section 4.2). For each situation, the predicted class is then provided together with a sentence describing the emotion type, emotion intensity and gender to the grounding algorithm (Algorithm 1). Before the actual grounding procedure, auxiliary words, which are words that have no corresponding concrete representations, are automatically detected and removed through Algorithm 2, which is a slightly modified version of the algorithm proposed in [38] and marks words that occur more than twice as often as the most often occurring concrete representation as auxiliary words. Afterwards, the set of word–concrete-representation pairs (*WCRPS*) and the set of concrete representation pairs (*CRWPS*) are updated based on the words and concrete representations encountered in the current situation. The former contains, for all previously encountered words, a set of concrete representations they co-occurred with as well as the corresponding co-occurrence counts, while the latter contains, for all previously encountered concrete representations, a set of words they co-occurred with as well as the corresponding co-occurrence counts. The word–concrete-representation (*WCRP*) and concrete-representation–word (*CRWP*) pairs that occurred the most based on the updated WCRPS and CRWPS are then added to the set of grounded words (*GW*) and concrete representations (*GCR*), respectively. To enable the algorithm to ground synonyms and homonyms, the words and concrete representations that were part of the highest WCRP and CRWP can be used again during future iterations, because these require that multiple words are able to be mapped to the same concrete representation and vice versa. Finally, the sets of grounded words and concrete representations are combined. The described grounding procedure is illustrated by Algorithm 1.

---

**Algorithm 1** The grounding procedure takes as input all words (*W*) and concrete representations (*CR*) of the current situation, the sets of all previously obtained word-concrete representation (*WCRPS*) and concrete representation-word (*CRWPS*) pairs, and the set of auxiliary words (*AW*) and returns the sets of grounded words (*GW*) and grounded concrete representations (*GCR*).

---

1:  **procedure** GROUNDING(*W*, *CR*, *WCRPS*, *CRWPS*, *AW*)

2:      Update *AW* (Algorithm 2) and remove *AW* from *W*

3:      Update *WCRPS* and *CRWPS* using *W* and *CR*

4:      **for** $j = 1$ to *word_number* **do**

5:          Save highest *WCRP* to *GW*

6:      **end for**

7:      **for** $j = 1$ to *concrete_representation_number* **do**

8:          Save highest *CRWP* to *GCR*

9:      **end for**

10:     **return** $GW \cup GCR$

11: **end procedure**

---

**Algorithm 2** The auxiliary word detection procedure takes as input the sets of word and concrete representation occurrences (*WO* and *CRO*), and the set of previously detected auxiliary words (*AW*) and returns an updated AW.

---

1:  **procedure** AUXILIARY WORD DETECTION(*WO*, *CRO*, *AW*)

2:      **for** word, occurrence in *WO* **do**

3:          **if** $occurrence > max(CRO) * 2$ **then**

4:              Add word to *AW*

5:          **end if**

6:      **end for**

7:      **return** *AW*

8: **end procedure**

---

## 5. Baseline Framework

The baseline framework uses the same percept extraction and classification components as the proposed framework (Sections 4.1 and 4.2), while it uses a probabilistic model to ground words through their corresponding concrete representations. The latter is described in detail in this section.

The probabilistic learning model is based on the model used in [19]. The model has been chosen as a baseline because similar models have previously been employed in many different grounding scenarios to ground a variety of percepts, such as shapes, colors, actions, or spatial relations [18–21]. In the model (Figure 2), the observed state $w_i$ represents word indices, i.e., each individual word is represented by a different integer. The following two example sentences illustrate the representation of words through word indices: (the, **1**) (man, **2**) (is, **3**) (very, **4**) (happy, **5**) and (the, **1**) (woman, **6**) (is, **3**) (really, **7**) (cheerful, **8**), where the bold numbers indicate word indices. Although "very" and "really" as well as "happy" and "cheerful" are synonyms in the context of this study (Table 2), they are represented by different word indices. The observed state $t$ represents the type of emotion, $s$ represents the strength or intensity of the emotions and $g$ represents genders.

Table 3 provides a summary of the definitions of the learning model parameters. The corresponding probability distributions, i.e., $w_i$, $\theta_{m,Z_{L_1}}$, $\phi_{t_{K_1}}$, $\phi_{s_{K_2}}$, $\phi_{g_{K_3}}$, $\pi_w$, $\pi_t$, $\pi_s$, $\pi_g$, $m_i$, $Z_t$, $Z_s$, $Z_g$, $t$, $s$, and $g$, which characterize the different modalities in the graphical model, are defined in Equation (1), where *GIW* denotes a Gaussian Inverse-Wishart distribution, and *N* denotes a multivariate Gaussian distribution. Gaussian distributions are used for $t$, $s$, and $g$ because concrete representations are represented by one-hot encoded vectors.

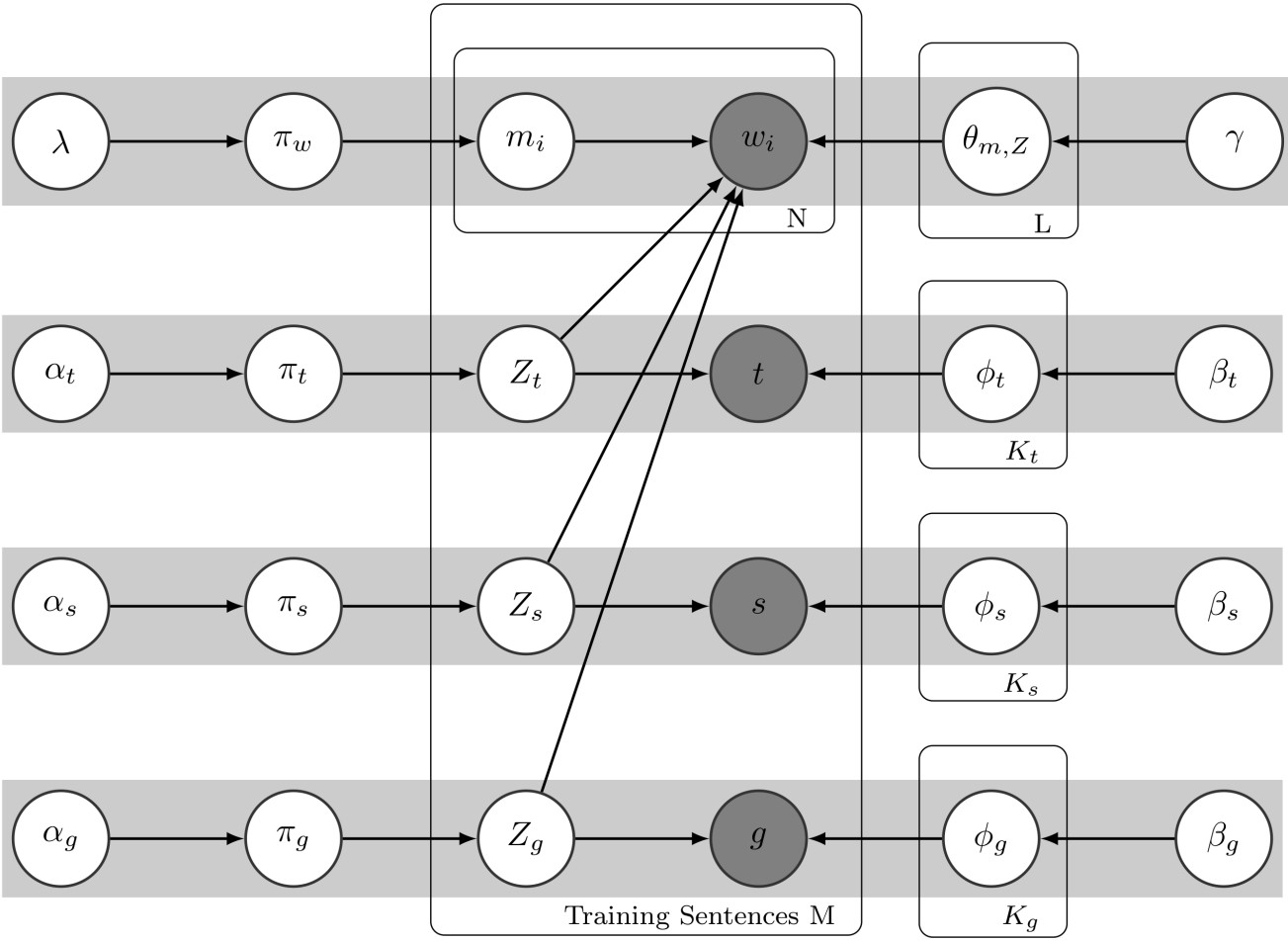

**Figure 2.** Graphical representation of the probabilistic model. Indices $i$, $t$, $s$, and $g$ denote the order of words, emotion types, emotion strengths, and genders, respectively.

**Table 2.** Overview of all concepts with their corresponding synonyms and concrete representation number (CR#) according to Figure 7.

| Type | Concept | Synonyms | CR# |
|---|---|---|---|
| Emotion Type | Happiness | happy, cheerful | 1 |
| | Sadness | sad, sorrowful | 2 |
| | Anger | angry, furious | 3 |
| | Neutral | neutral, fine | 4 |
| | Surprise | surprised, startled | 5 |
| | Fear | afraid, scared | 6 |
| | Disgust | disgusted, appalled | 7 |

**Table 2.** *Cont.*

| Type | Concept | Synonyms | CR# |
|---|---|---|---|
| Emotion Intensity | Normal [1] | slightly, lightly | 8 |
| | Strong | very, really | 9 |
| Gender | Female | she, woman | 10 |
| | Male | he, man | 11 |
| Auxiliary word | - | the, is | 0 |

[1] Since emotion intensities are binary, i.e., strong or normal, normal was seen as synonymous to weak, which made it easier to find appropriate words.

**Table 3.** Definitions of the learning parameters in the graphical model.

| Parameter | Definition |
|---|---|
| $\lambda$ | Hyperparameter of the distribution $\pi_w$ |
| $\alpha_t, \alpha_s, \alpha_g$ | Hyperparameters of the distributions $\pi_t, \pi_s$ and $\pi_g$ |
| $m_i$ | Modality index of each word (modality index $\in$ {Type, Strength, Gender, AW}) |
| $Z_t, Z_s, Z_g$ | Indices of type, strength and gender distributions |
| $w_i$ | Word indices |
| $t, s, g$ | Observed states representing types, strengths and genders |
| $\gamma$ | Hyperparameter of the distribution $\theta_{m,Z}$ |
| $\beta_t, \beta_s, \beta_g$ | Hyperparameters of the distributions $\phi_t, \phi_s$ and $\phi_g$ |
| $\theta_{m,Z}$ | Word distribution over modalities |

$$
\begin{cases}
w_i & \sim & Cat(\theta_{m_i, Z_{m_i}}) \\
\theta_{m, Z_{L_1}} & \sim & Dir(\gamma) \quad, \quad L_1 = (1, ..., L) \\
\phi_{t_{K_1}} & \sim & GIW(\beta_t), \quad K_1 = (1, ..., K_t) \\
\phi_{s_{K_2}} & \sim & GIW(\beta_s), \quad K_2 = (1, ..., K_s) \\
\phi_{g_{K_3}} & \sim & GIW(\beta_g), \quad K_3 = (1, ..., K_g) \\
\pi_w & \sim & Dir(\lambda) \\
\pi_t & \sim & Dir(\alpha_t) \\
\pi_s & \sim & Dir(\alpha_s) \\
\pi_g & \sim & Dir(\alpha_g) \\
m_i & \sim & Cat(\pi_w) \\
Z_t & \sim & Cat(\pi_t) \\
Z_s & \sim & Cat(\pi_s) \\
Z_g & \sim & Cat(\pi_g) \\
t & \sim & N(\phi_{Z_t}) \\
s & \sim & N(\phi_{Z_s}) \\
g & \sim & N(\phi_{Z_g})
\end{cases}
\tag{1}
$$

The latent variables of the Bayesian learning model are inferred using the Gibbs sampling algorithm [44] (Algorithm 3), which repeatedly samples from and updates the posterior distributions (Equation (2)). Distributions were sampled for 100 iterations, after which convergence was achieved.

$$
\begin{cases}
\phi_t & \sim \quad P(\phi_t|t,\beta_t) \\
\phi_s & \sim \quad P(\phi_s|s,\beta_s) \\
\phi_g & \sim \quad P(\phi_g|g,\beta_g) \\
\pi_w & \sim \quad P(\pi_w|\lambda,m) \\
\pi_t & \sim \quad P(\pi_t|\alpha_t,Z_t) \\
\pi_s & \sim \quad P(\pi_s|\alpha_s,Z_s) \\
\pi_g & \sim \quad P(\pi_g|\alpha_g,Z_g) \\
Z_t & \sim \quad P(Z_t|t,\pi_t,w) \\
Z_s & \sim \quad P(Z_s|s,\pi_s,w) \\
Z_g & \sim \quad P(Z_g|g,\pi_g,w) \\
\theta_{m,Z} & \sim \quad P(\theta_{m,Z}|m,Z_t,Z_s,Z_g,\gamma,w) \\
m_i & \sim \quad P(m_i|\theta_{m,Z},Z_t,Z_s,Z_g,\pi_w,w_i)
\end{cases}
\tag{2}
$$

---

**Algorithm 3** Inference of the model's latent variables. *iter_num* was set to 100.

---

1: **procedure** GIBBS SAMPLING($W$, $P$, $WP$, $AW$)

2:     Initialization of $\theta, \phi_t, \phi_s, \phi_g, \pi_w, \pi_t, \pi_s, \pi_g, Z_t, Z_s, Z_g, m_i$

3:     **for** $i = 1$ to *iter_num* **do**

4:         Equation (2)

5:     **end for**

6:     **return** $\theta, \phi_t, \phi_s, \phi_g, \pi_w, \pi_t, \pi_s, \pi_g, Z_t, Z_s, Z_g, m_i$

7: **end procedure**

---

## 6. Experimental Setup

The proposed framework (Section 4) is evaluated through human–agent interactions in simulated situations created using the RAVDESS dataset [40], which consists of frontal face pose videos of twelve female and twelve male north American actors and actresses, who speak and sing two lexically matched sentences while expressing six basic emotions, i.e., happiness, surprise, fear, disgust, sadness, and anger [45], plus calmness and neutral, through their voice and facial expressions. In this study, only the speaking records of the six basic emotions and neutral are used. In addition to the expressed emotions, all videos in which one of the six basic emotions is expressed also come with labels indicating the intensities of the expressed emotions, i.e., normal or strong, which are used in this study to train the emotion intensity recognition model (Section 4.2). Since the videos of eighteen actors and actresses are used to train the percept classifiers (Section 4.2), only the videos of six actors and actresses are used to create situations for the simulated human–agent interactions, leading to a total of 312 situations, i.e., for each person, eight videos per basic emotion (four for each intensity level) and four videos for neutral (only one intensity level). Each situation is created according to the following procedure:

1. The video representing the current situation is given to OpenEAR, which extracts 384 features (Section 4.1);
2. 156 (MFCC and PCM RMS) of the 384 features are provided as input to the employed deep neural networks to determine the concrete representations of the expressed emotion, its intensity and the gender of the person expressing it (Section 4.2);
3. The concrete representations are provided to the agent together with a sentence describing the emotion type, intensity and gender of the person in the video, e.g., "She is very angry.";
4. The agent uses cross-situational learning to ground words through corresponding concrete representations (Sections 4.3 and 5).

Each sentence has the following structure: "(the) *gender* is (*emotion intensity*) *emotion type*", where *gender*, *emotion intensity* and *emotion type* are replaced by one of their corre-

sponding synonyms (Table 2). If the emotion type is "neutral", the intensity is always normal; thus, the sentence does not contain a word describing the intensity of the emotion and no corresponding concrete representation is provided to the agent. Additionally, if the gender is described by a noun, i.e., "woman" or "man", it is preceded by the article "the". Since the words used to describe a situation are randomly chosen from the available synonyms, how often each word occurs during training and testing varies, e.g., "cheerful" appears nearly twice as many times during training and testing as its synonym "happy" (Figure 3). Ten different interaction sequences, for which the order of the situations was randomly changed, are used to evaluate the grounding frameworks to ensure that the obtained results are independent of the specific order in which situations are encountered. The proposed framework receives situations one after the other as if it is processing the data in real-time during the interaction, while the baseline framework requires all sentences and corresponding concrete representations of the training situations to be provided at the same time. Therefore, two different cases are evaluated. First, the case in which all situations are used for training and testing, because this allows the proposed framework to continuously learn, while it is an unrealistic case for the baseline framework because it is very unlikely that all situations have already been encountered during training. Second, only 60% of the situations are used for training, which is more realistic for the baseline framework, while it adds an unnecessary limitation to the proposed framework by deactivating its learning mechanism for 40% of the situations, although it does not require an explicit training phase.

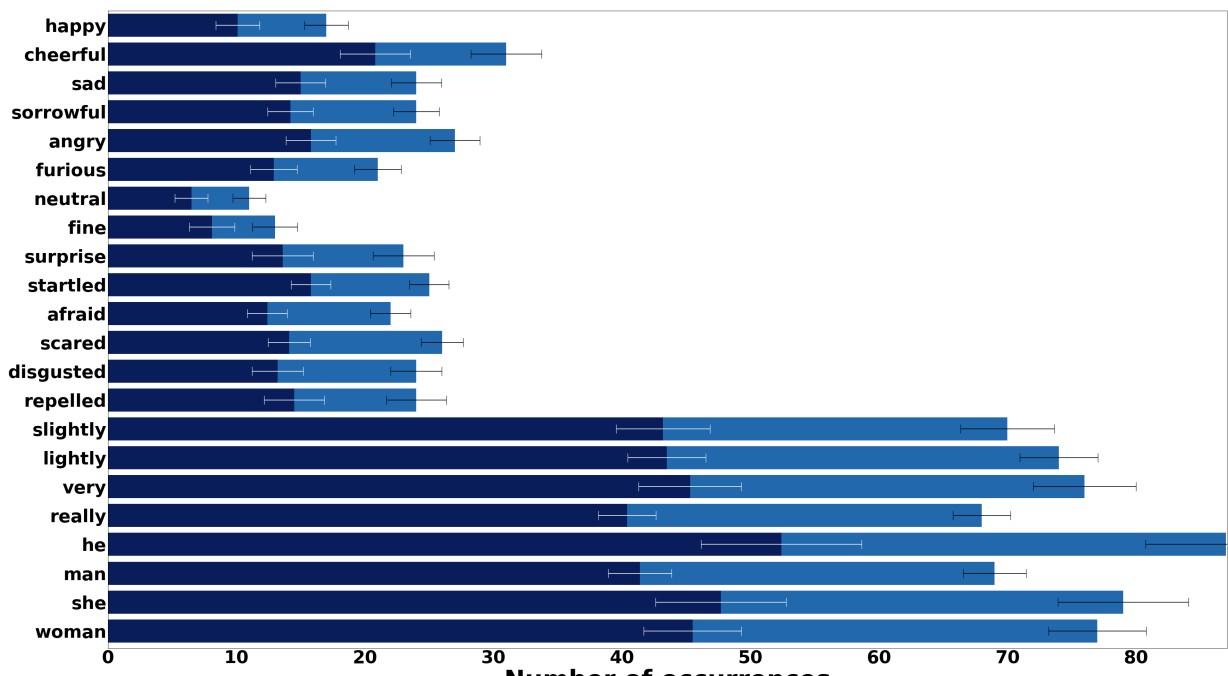

**Figure 3.** Word occurrences for all words except auxiliary words. The dark blue part of the bars shows the mean number of occurrences during training and the bright blue part shows the mean number of occurrences during testing.

## 7. Results and Discussion

The proposed cross-situational learning based framework (Section 4) is evaluated through a simulated human–agent interaction scenario (Section 6) and the obtained grounding results are compared to the groundings achieved by an unsupervised Bayesian grounding framework (Section 5). Since the same percept extraction and classification components (Sections 4.1 and 4.2) are used for both frameworks, any difference in grounding performance can only be due to the different grounding algorithms described in Sections 4.3 and 5.

Figure 4 shows how the mean number of correct and false mappings obtained by the proposed framework changes over all 312 situations. It shows two different cases, which

differ regarding the concrete representations used for emotion types, i.e., for the first case (TPRE), the predicted concrete representations are used, while for the second case (TPER), perfect concrete representations are used to investigate the effect of the accuracy of the concrete representations on the grounding performance. For TPRE, represented by continuous lines, the number of correct mappings quickly increases from zero to about twelve mappings for the first 20 situations, and continuous to increase more slowly afterwards to 15 mappings, while the number of false mappings starts with about six mappings and increases over the course of 45 situations to 15 mappings, after which it slowly decreases to 13 mappings. The main reason for the large number of false mappings is that the concrete representations used for emotion types are highly inaccurate, with an accuracy of 59.6%, while, at the same time, 60% of the employed words refer to them. This assumption is confirmed when looking at TPER, represented by the dashed line, which shows the number of correct and false mappings when perfect concrete representations are used for emotion types, while the predicted ones are still used for the other two modalities, i.e., emotion intensity and gender. For TPER, the proposed framework obtains 17 and 20 correct mappings within the first 20 and 45 situations, respectively. If the framework is only allowed to learn during 60% of the situations, it obtains 21 correct mappings, while it obtains one more mapping, i.e., 22, if it continues learning for the remaining situations. In contrast, the number of false mappings increases slightly from five to seven from the first to the second situation, stays stable for about eight situations and decreases then continuously to two mappings after 60% of the situations have been encountered and one mapping after all situations have been encountered. Both cases together illustrate that the proposed grounding algorithm depends on the accuracy of the obtained concrete representations; however, it does not require perfectly accurate representations because it is able to obtain all correct mappings for the second case, although the concrete representations for emotion intensities and genders only have accuracies of 73.5% and 89.8%, respectively.

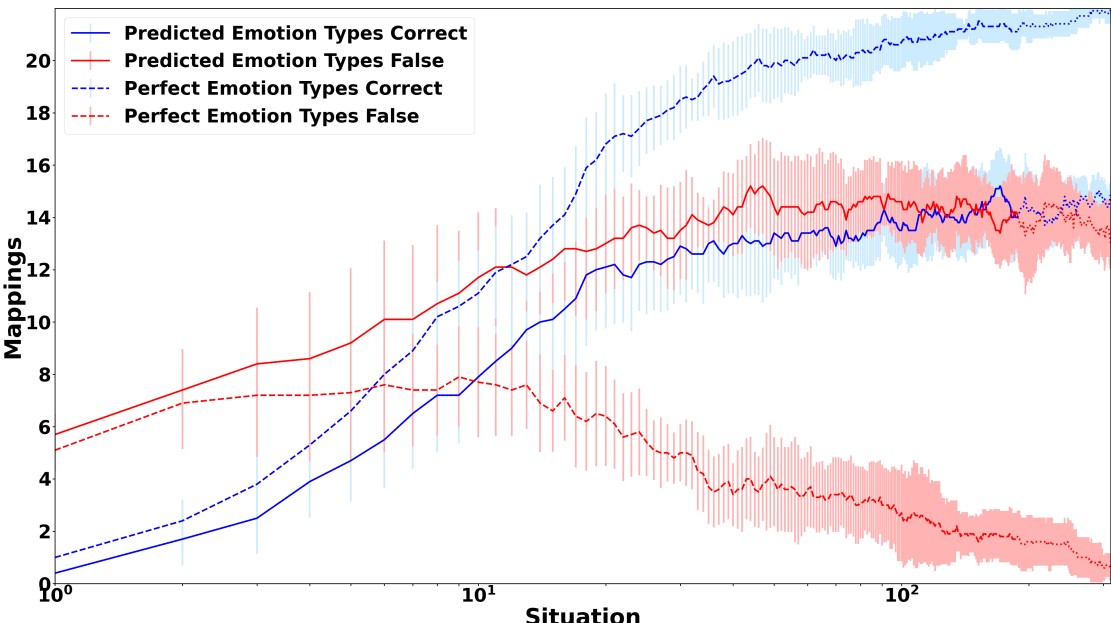

**Figure 4.** Mean number and standard deviation of correct and false mappings obtained by the proposed model over all 312 situations. The continues line represents the results when the predicted concrete representations are used for all modalities, while the dashed line represents the results when perfect concrete representations are used for emotion types to investigate the influence of the concrete representation accuracy on the grounding performance of the proposed model. For all lines, the dotted parts only occur when all situations are used for training.

The figure also illustrates the online grounding capability and transparency of the proposed grounding algorithm because it updates its mappings for every new encountered

situation and allows, at any time, to check through which concrete representation a word is grounded. The latter becomes important when the model is used in real human–agent interaction to understand and debug the agent's actions, especially in cases where they might have been inappropriate. Since the baseline model requires an explicit training phase, no similar figure can be obtained. Thus, to compare the two models, the mappings of the proposed model are extracted after 187 and 312 situations depending on the used train/test split. In this study, two different train/test splits are used. For the first split (TTS60), only 60% of the situations are used for training and the remaining 40% for testing to investigate how well the models perform for unseen situations. Figure 3 provides an overview about the average occurrence of each word in the train and test sets. Applying the learning mechanisms of the proposed model only for the first 187 situations is both unnecessary and unrealistic because it is able to learn in an online manner and does not require an explicit training phase; however, it has been done out of fairness to the baseline model because the latter sees also only 60% of the situations during training. In contrast, for the second split (TTS100,) all situations are used for training and testing to ensure that the proposed model can learn continuously, while providing an unrealistic benefit to the baseline model because it is very unlikely that it would encounter all situations already during the offline training phase.

Figure 5 shows the accuracies for each modality for both models and test splits, as well as the percentage of sentences for which all words were correctly grounded. Additionally, it also illustrates the influence of the accuracy of the employed concrete representations by showing both the results for the predicted concrete representations of emotion types (Figure 5a) and when using perfect concrete representations for emotion types (Figure 5b). The proposed model achieves a higher accuracy than the baseline model in all cases, i.e., for all modalities, train/test splits and both concrete representations of emotion types, except for emotion types, when the predicted concrete representations are used and all situations are encountered during training. In fact, for genders, the proposed model achieves perfect grounding due to the high accuracy of the corresponding concrete representations, i.e., 89.8%. The figure also confirms the results in Figure 4 that the grounding accuracy improves with the number of encountered situations, which seems intuitive but is not necessarily the case, as shown by the results obtained for the baseline model, i.e., the latter obtained less accurate groundings for most modalities when using all situations for training and testing due to the larger number of situations in the test set. For the baseline model, using perfect concrete representations for emotion types increases the accuracy of the groundings obtained for emotion types and genders as well as the accuracy of auxiliary words, although the accuracy of the latter two only increases for TTS100, while the accuracy of the emotion intensity groundings decreases independent of the number of situations encountered during training.

Although the accuracies provide a good overview of how accurate the groundings for each modality are, they do not provide any details about the wrong groundings or the accuracy of the groundings obtained for individual words. Therefore, Figure 6 shows the confusion matrices for all words and modalities, which illustrate how often each word was grounded through the different modalities and highlight two interesting points. First, both models show a high confusion for emotion types, i.e., all of them have non-zero probabilities to be mapped to concrete representations representing emotion intensities or genders, due to the low accuracy of the corresponding concrete representations for TTS60. The confusion decreases for TSS100, in which case most words converge to one modality for the proposed model, i.e., only "happy" and "sad" are still confused as a gender or emotion intensity, respectively. However, this does not lead to a substantial increase in grounding accuracy for emotion types because some words, e.g., "surprise" and "afraid", converge to the wrong modality so that the probability to be mapped to a concrete representation of an emotion type decreases to zero. Figure 7 shows confusion matrices of words and different concrete representations, thereby allowing to investigate whether the concrete representation a word is grounded through is correct, which might

not be the case if there is a high confusion between concrete representations of the same modality. The fourth column, representing the emotion type neutral, is very noticeable in Figure 7a,c,d because both models do not map any word to it, except for the proposed model and TTS60 (Figure 7b); however, even in the latter case, the probability that the word "fine" gets mapped to it is very low because most of the time it is mapped to the concrete representation of the concept male (column 11). Otherwise, the results show that, for the proposed model, the confusion is normally across modalities and not between concrete representations of the same modality. In contrast, the baseline model shows strong confusions between concrete representations of the same modality, e.g., for TTS60 "happy" and "disgusted" are more often grounded through anger than happiness and disgust, respectively.

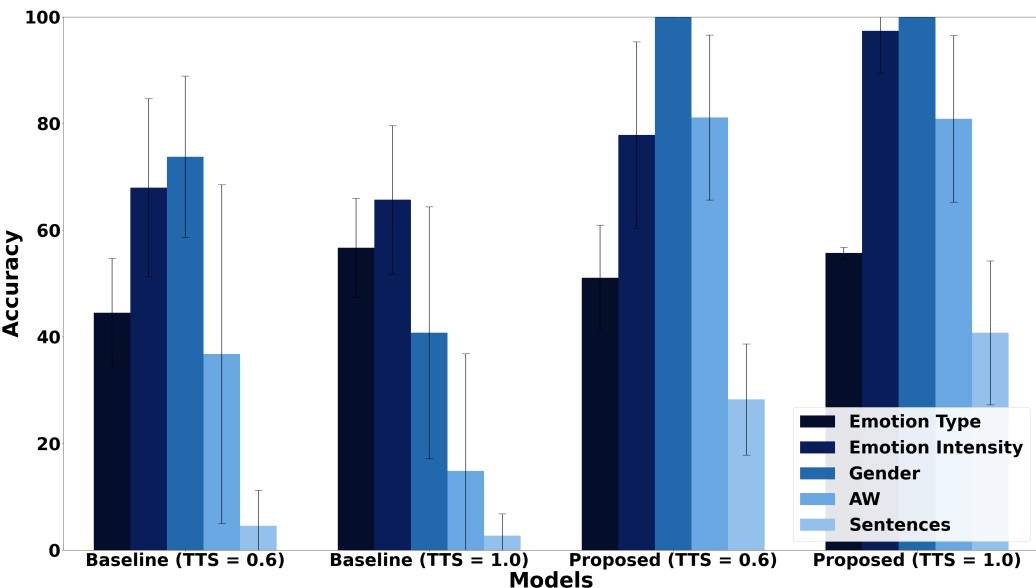

(**a**) Results when the predicted concrete representations are used for all modalities.

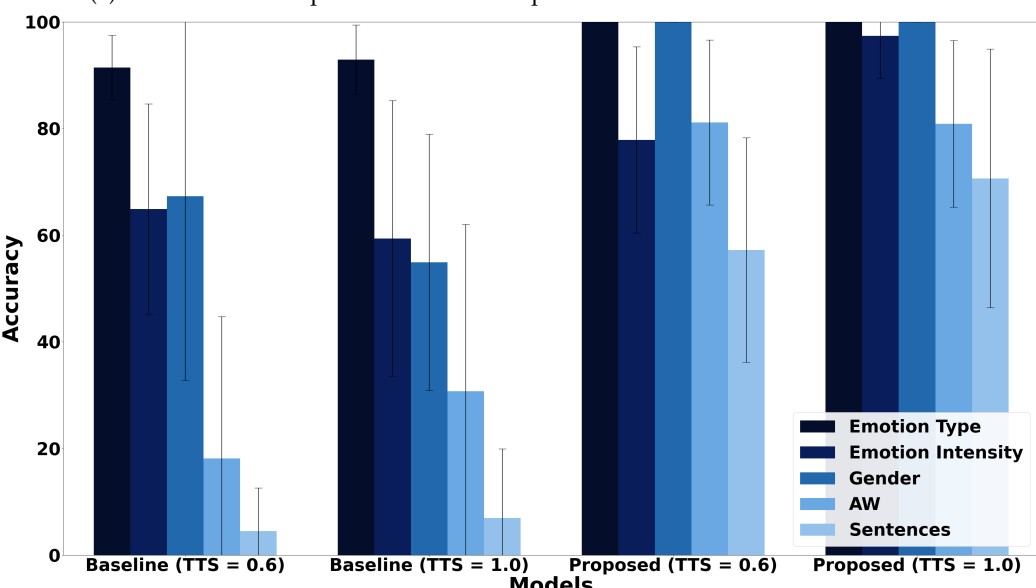

(**b**) Results when perfect concrete representations are used for emotion types.

**Figure 5.** Mean grounding accuracy results and corresponding standard deviations for both grounding models, all modalities and both training/test splits (TTS). Additionally, the percentage of sentences for which all words were correctly grounded is shown.

When considering the deployability of the proposed and the baseline framework, it is important to also analyse the required computational resources. The grounding experiments have been conducted on a system with Ubuntu 16.04, i7-6920HQ CPU, octa core with 2.90 GHz each, and 32 GB RAM. However, it is important to note that both frameworks are only utilizing a single core; thus, the same processing times would be achieved with a single core, if no other computationally expensive processes are running at the same time. The average time it took the proposed framework to process a new situation and update its mappings was 3 ms, while the inference time was only 56 µs. In contrast, one Gibbs sampling iteration of the baseline model took 647 ms. Since 100 iterations were used, the average training time (averaged across all 10 runs) for the baseline model was 65 s for all 320 situations, while the inference time was on average 7.45 ms for each situation. These results confirm that both framework can be used for real-time grounding applications, while only the proposed framework can be used for dynamic environments that require frequent updates of the models because the baseline framework requires already more than one minute to train on a relatively small number of situations, which also needs to be done in advance and is, therefore, not possible after deployment.

Overall, the evaluation shows that the proposed model outperforms the baseline in terms of auxiliary word detection and grounding accuracy as well as its abilities to learn continuously without requiring explicit training. The latter does not only make it more applicable for real-world scenarios but also more transparent, because it is possible to observe how a new situation influences the obtained groundings.

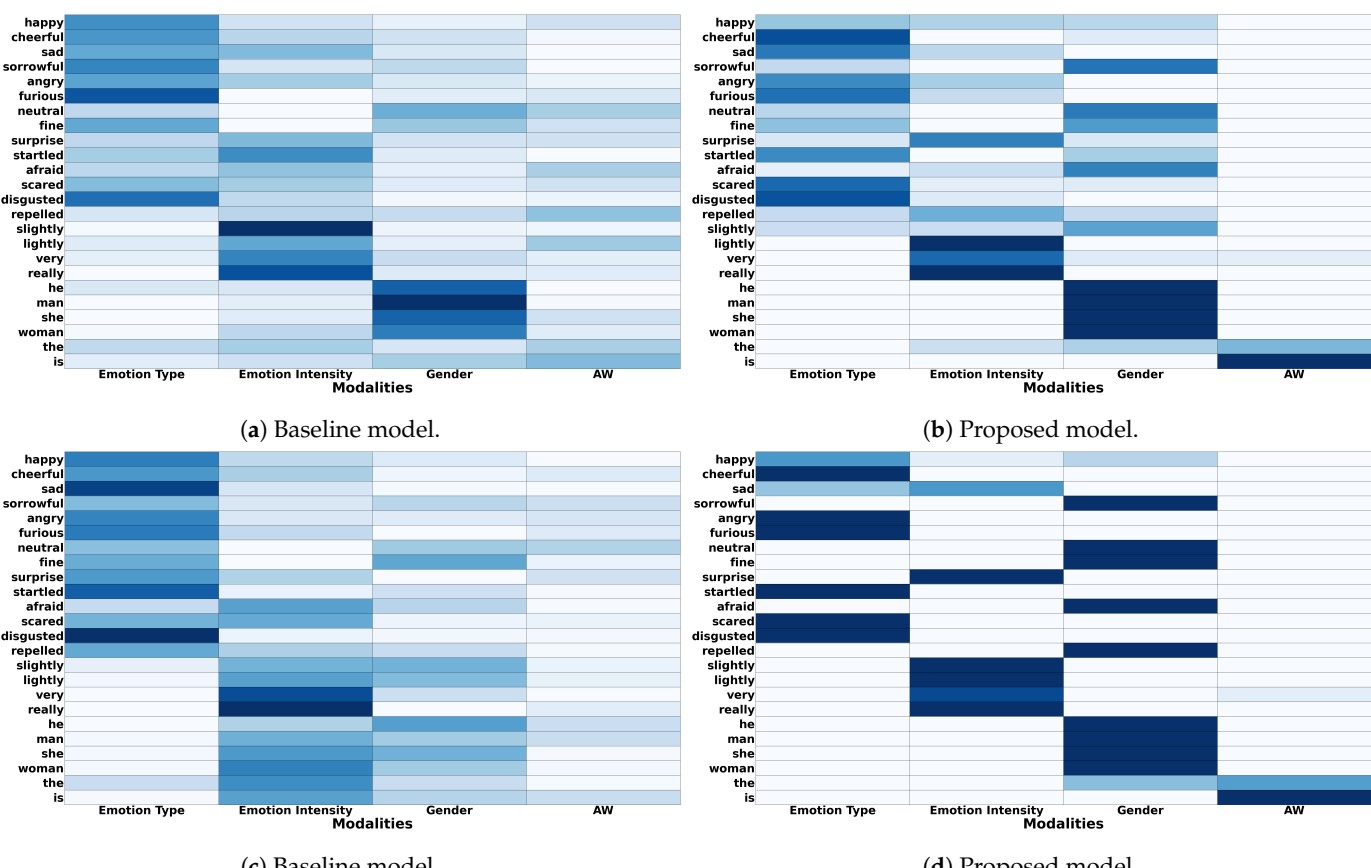

**Figure 6.** Confusion matrices showing how often each word was grounded on average, i.e., over ten runs, through which modality when using predicted concrete representations for all modalities. (**a**,**b**) show the results when only 60% of the situations are used for training, while (**c**,**d**) show the results when all situations are used for training and testing.

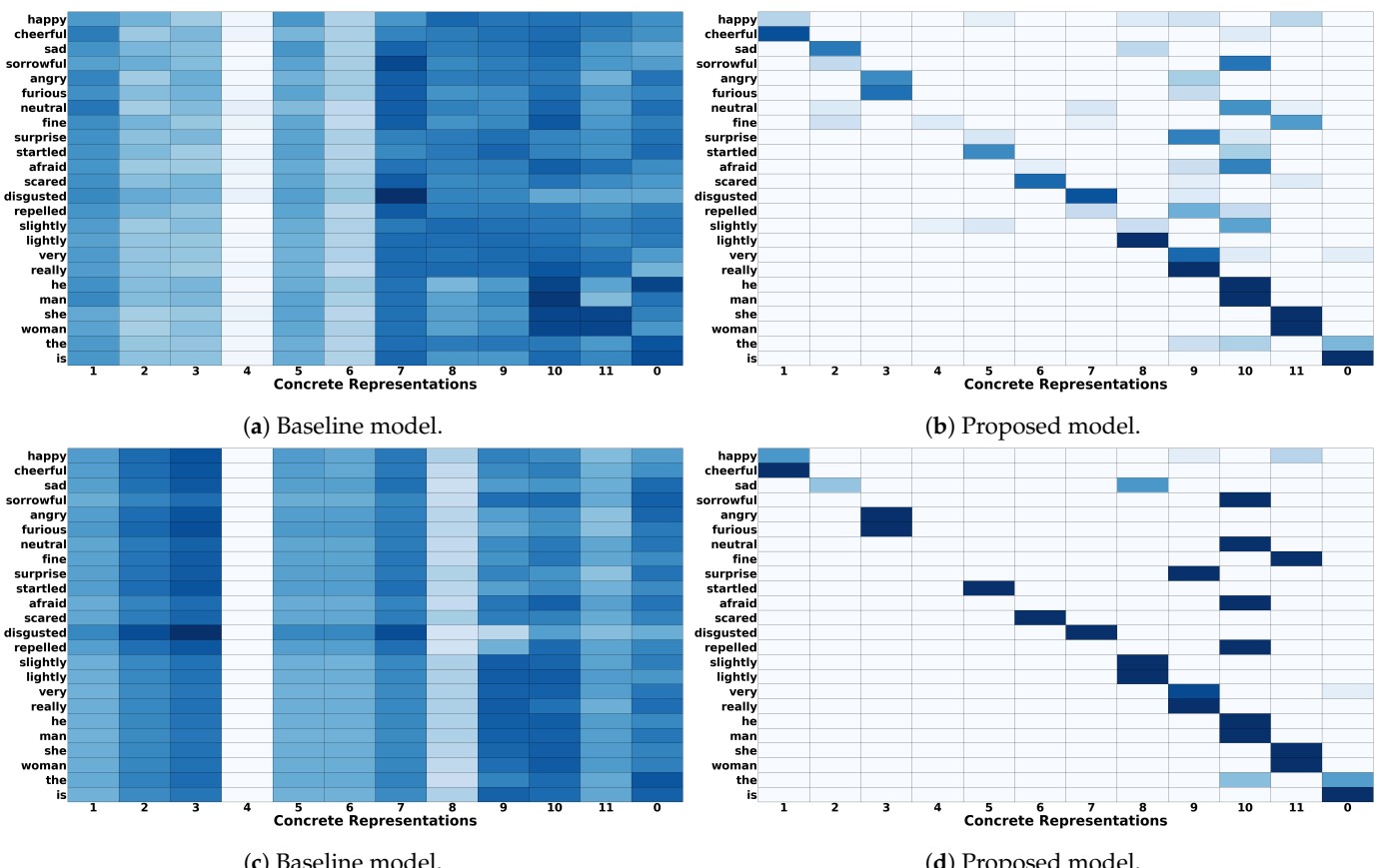

**Figure 7.** Confusion matrices of words over different concrete representations for all ten situation sequences when using predicted concrete representations for all modalities. (**a**,**b**) show the results when only 60% of the situations are used for training, while (**c**,**d**) show the results when all situations are used for training and testing.

## 8. Conclusions and Future Work

This paper investigated whether the proposed unsupervised online grounding framework is able to ground abstract concepts, like emotion types, emotion intensities and genders, during simulated human–agent interactions. Percepts were converted to concrete representations through deep neural networks that received as input audio features extracted via OpenEAR from videos.

The results showed that the framework is able to identify auxiliary words and ground non-auxiliary words, including synonyms, through their corresponding emotion types, emotion intensities and genders. Additionally, the proposed framework outperformed the baseline model in terms of the accuracy of the obtained groundings, as well as its ability to learn new groundings and continuously update existing groundings during interactions with other agents and the environment, which is essential when considering real-world deployment. Furthermore, the framework is also more transparent, due to the creation of explicit mappings from words to concrete representations.

In future work, we will investigate whether the framework can be used to ground homonyms, i.e., concrete representations that can be referred to by the same word. Furthermore, we will investigate whether the framework can ground emotion types, intensities and genders, if multiple people are present in a video. Finally, we are planning to integrate the framework with a knowledge representation to explore the utilization of abstract knowledge to increase the sample-efficiency of the grounding mechanism as well as the accuracy of the obtained groundings, and enable agents to reason about the world with the help of an abstract but grounded world model.

**Author Contributions:** Conceptualization, O.R.; methodology, E.B. and O.R.; software, E.B. and O.R.; validation, E.B. and O.R.; formal analysis, O.R.; investigation, E.B. and O.R.; resources, E.B. and O.R.; data curation, E.B. and O.R.; writing—original draft preparation, O.R.; writing—review and editing, E.B. and O.R.; visualization, E.B. and O.R.; supervision, O.R.; project administration, O.R.; funding acquisition, O.R. All authors have read and agreed to the published version of the manuscript.

**Funding:** The APC was funded by the Flemish Government under the program Onderzoeksprogramma Artificiele Intelligentie (AI) Vlaanderen.

**Data Availability Statement:** The used perceptual data is available at https://zenodo.org/record/1188976, accessed on 17 April 2021, while the used sentences are available on request from the corresponding author.

**Conflicts of Interest:** The authors declare no conflict of interest.

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
