# Peer review of "Unsupervised Online Grounding for Social Robots"

_robotics, doi:10.3390/robotics10020066_

Round 1

Reviewer 1 Report

Dear authors,

I enjoyed reading your paper. It was well-written and interesting, and I find your research question on grounding more abstract concepts a highly relevant one to the field of human-robot interaction. Below, I have written a few comments and questions of which I hope that they will be useful in further improving your manuscript and making it accessible to a wider audience.

I mostly struggle with understanding your results section. I understand that you compare two different versions of your framework (using 60% vs 100% of the situation in the training process) to the baseline model. However, in your results section, it is not always clear to me which model you are discussing. It would be useful if you could provide a bit more structure to the reader by clearly explaining which models you are comparing, and by consistently using names for your models throughout the manuscript.

I have a question regarding your choice to select gender as an abstract concept. How abstract is gender really? I understand that emotion types and intensities are more abstract than concrete concepts of which artificial agents receive input through their sensors, but gender is somewhat in between. Could you elaborate on this choice?

In your discussion and conclusion, I would like to read a bit more about the contribution of your research to the general field. You briefly comment that your framework is more applicable for the real world, but I think you could expand on this. In line with that, I find your ideas for future work interesting, but mostly relevant for the short term. What are your long-term ideas about the further development of your framework? Do you have an ultimate goal to which you are working?

One comment regarding your style of writing: some sentence are quite long, which makes them a bit difficult to understand (e.g., line 26, 65, 82 or 102). You could consider breaking them up into multiple sentences, or put “i.e. parts” in brackets.

I hope my comments were useful and I wish you the best of luck in your future research!

Author Response

Dear reviewer,   thank you for your comments and questions.   Please find our replies and clarifications below. For some of the comments we also made changes in the manuscript, which we have highlighted in yellow.   Best regards, The authors

I mostly struggle with understanding your results section. I understand that you compare two different versions of your framework (using 60% vs 100% of the situation in the training process) to the baseline model. However, in your results section, it is not always clear to me which model you are discussing. It would be useful if you could provide a bit more structure to the reader by clearly explaining which models you are comparing, and by consistently using names for your models throughout the manuscript.

We added names for the different evaluation cases, i.e. TPRE and TPER refer to the cases when predicted and perfect concrete representations are used for emotion types, respectively, while TTS60 and TS100 refer to the cases when only 60% and 100% of the situations are encountered during training, respectively.

I have a question regarding your choice to select gender as an abstract concept. How abstract is gender really? I understand that emotion types and intensities are more abstract than concrete concepts of which artificial agents receive input through their sensors, but gender is somewhat in between. Could you elaborate on this choice?

It is true that emotion types and intensities can be seen as more abstract than gender because the former are only temporary conditions, while the latter is a permanent state (most humans never change their gender during their life).

However, gender is still more abstract than the shape or color of an object because the latter can be directly described by specific perceptual features. For example, the color of an object can be described by the RGB values of each pixel belonging to the target object and the shape of an object can be described by its width and height. In contrast, a gender cannot reliably described by a small set of visual or audio features but the set of appropriate features is context dependent and more difficult to obtain.

In your discussion and conclusion, I would like to read a bit more about the contribution of your research to the general field. You briefly comment that your framework is more applicable for the real world, but I think you could expand on this. In line with that, I find your ideas for future work interesting, but mostly relevant for the short term. What are your long-term ideas about the further development of your framework? Do you have an ultimate goal to which you are working?

The "ultimate goal" is to enable natural and efficient human-agent interactions in human-centered complex environments, like private homes, schools, or retirement homes, by equipping agents with the necessary mechanisms to "understand" natural language and adapt their behavior based on the preferences of the humans they are interacting with. Since it is impossible to learn all possible groundings in an offline training phase, it is essential that the employed grounding mechanisms work online, which is not the case for many frameworks as explained in the paper. One of the long-term goals is to integrate the grounding framework with a knowledge representation so that the agent can reason about the world with the help of an abstract world model. The idea is that the grounding framework will be used to ensure that the abstract model is connected to the real world so that any plan generated with the help of the abstract model is in the end also executable in the real world.

The presented framework is, of course, just a first step towards that goal. However, with that goal in mind we designed it to be able to learn in an online fashion and also to represent word-concrete representation mappings in an explicit manner because this will simplify the integration with a knowledge representation. We revised and extended the conclusion to make the contribution of the presented research more clear and explain why it is more applicable for the real world than other previously proposed grounding frameworks. We also added an additional paragraph to the results section to compare the processing speed of both frameworks to further illustrate that the proposed framework is more applicable for the real world than other grounding frameworks. Since the timing analysis was a suggestion of the second reviewer the paragraph has been highlighted in green.

One comment regarding your style of writing: some sentence are quite long, which makes them a bit difficult to understand (e.g., line 26, 65, 82 or 102). You could consider breaking them up into multiple sentences, or put “i.e. parts” in brackets.

We went through the whole paper and broke long sentences where possible into shorter ones, e.g. lines 65, 68, 83, 101, and 332.

Reviewer 2 Report

In this paper, Cross-situational learning is applied in order for an agent (a social robot) to be able to understand the emotions, the
emotional intensity and the gender of humans interacting with it. 
The technical presentation and the manuscript organization are satisfactory. 
After studying the manuscript and the related references the following comments are stated:

1) The used term "Percept Extraction" seems to be not appropriate. In fact, what is proposed by the authors in this processing step is "Feature Extraction". At this point you do not know what is the meaning of the feature representation, it is abstract.
2) In Lines 133-134 the authors claimed that they selected 156 out of 328 features because they provided the best classification performance.
However, the authors did not describe the method they applied for selecting the features.
3) It is not clear if the same 156 features are used in three DL models. If they used the same features, then it is questionable the ability of these features to describe different outcomes (emotion state, emotion intensity, and gender). 
4) The authors wrongly used the term "detection" to describe the "emotion detection", "emotion intensity detection" and gender detection".
The proper term in these cases is "recognition". The authors recognize the emotional state, the emotional intensity, and the gender.
5) A complexity and timing performance analysis of the proposed methodology are not included in the manuscript. If we are
talking about a method that will be deployed on a social robot the real-time performance is a strong requirement. 
Recently, the following paper highlights the limitations of using such demanding AI algorithms for developing interaction skills
for social robots. Please the authors to provide information regarding this important aspect.

C.T. Recchiuto and A. Sgorbissa, "A Feasibility Study of Culture-Aware Cloud Services for Conversational Robots," IEEE Robotics and Automation Letters, vol. 5, no. 4, pp. 6559-6566, 2020. https://doi.org/10.1109/LRA.2020.3015461.

G.K. Sidiropoulos, C. Bazinas, C. Lytridis, G.A. Papakostas, V.G. Kaburlasos, , P. Kechayas, E. Kourampa, S.R. Katsi, C. Karatsioras
"Synergy of Intelligent Algorithms for Efficient Child-Robot Interaction in Special Education: A Feasibility Study,"
International Conference on Robotics in Education (RiE), pp. 98-105, 2020.https://doi.org/10.1007/978-3-030-67411-3_9

Overall, the paper extends the usage of Cross-situational learning in learning social norms and contributes to the development of more interactive social robots. However, the applicability of the proposed methodology in practice 
cannot be identified since relative information is not provided. 

Author Response

Dear reviewer,   thank you for your comments.   Please find our replies and clarifications below. For some of the comments we also made changes in the manuscript, which we have highlighted in green.   Best regards, The authors  

1) The used term "Percept Extraction" seems to be not appropriate. In fact, what is proposed by the authors in this processing step is "Feature Extraction". At this point you do not know what is the meaning of the feature representation, it is abstract.

Percepts are features obtained through the sensors of an agent, thus, "percept extraction" is in the context of the paper synonymous with "feature extraction". However, we agree that "percept extraction" might be less meaningful for many readers than "feature extraction", thus, we changed it to "perceptual feature extraction" to be both more general than "percept extraction" as well as more precise than just "feature extraction". Additionally, we also changed "percept classification" to "perceptual feature classification" because the corresponding model classifies perceptual features according to their concrete representations they belong to.

2 ) In Lines 133-134 the authors claimed that they selected 156 out of 328 features because they provided the best classification performance.

However, the authors did not describe the method they applied for selecting the features.

We performed an experimental evaluation of the available feature sets, i.e. we provided each individual feature set and different combinations of the feature sets as input to the employed architectures, and compared the mean accuracies obtained across five different runs. We added these details including a list of the used feature sets in the paper at the end of the perceptual feature extraction section for future readers.

3 ) It is not clear if the same 156 features are used in three DL models. If they used the same features, then it is questionable the ability of these features to describe different outcomes(emotion state, emotion intensity, and gender).

The same 156 features, i.e. MFCC and PCM RMS features, were used for all three models because they led to the best results for all of them (see reply to your previous comment). That the same features, i.e. MFCC and PCM RMS, provide the best results for all three classification tasks is not surprising and supported by the literature since many different studies have used MFCC features for a variety of tasks including speech recognition [1], speaker recognition [2], and stress detection [3].

[1] C. Ittichaichareon, S. Siwat, and Y. Thaweesak: Speech Recognition using MFCC. International Conference on Computer Graphics, Simulation and Modeling (ICGSM), 2012.

[2] Tiwari, Vibha: MFCC and its Applications in Speaker Recognition. In International Journal on Emerging Technologies, vol. 1 no. 1, pp. 19-22, 2010, ISSN: 0975-8364.

[3] M. Julião, J. Silva, A. Aguiar, H. Moniz, and F. Batista: Speech Features for Discriminating Stress Using Branch and Bound Wrapper Search. In: Sierra-Rodríguez JL., Leal JP., Simões A. (eds) Languages, Applications and Technologies. SLATE 2015. Communications in Computer and Information Science, vol. 563, Springer, Cham, 2015. https://doi.org/10.1007/978-3-319-27653-3_1

4 ) The authors wrongly used the term "detection" to describe the "emotion detection", "emotion intensity detection" and gender detection".

The proper term in these cases is "recognition". The authors recognize the emotional state, the emotional intensity, and the gender.

It is true that recognition is a more appropriate term for the described models, thus, we replaced the term "detection" with the term "recognition".

5 ) A complexity and timing performance analysis of the proposed methodology are not included in the manuscript. If we are talking about a method that will be deployed on a social robot the real-time performance is a strong requirement.

Recently, the following paper highlights the limitations of using such demanding AI algorithms for developing interaction skills for social robots. Please the authors to provide information regarding this important aspect.

C.T. Recchiuto and A. Sgorbissa, "A Feasibility Study of Culture-Aware Cloud Services for Conversational Robots," IEEE Robotics and Automation Letters, vol. 5, no. 4, pp. 6559-6566,2020. https://doi.org/10.1109/LRA.2020.3015461.

G.K. Sidiropoulos, C. Bazinas, C. Lytridis, G.A. Papakostas, V.G. Kaburlasos, P. Kechayas, E. Kourampa, S.R. Katsi, C. Karatsioras "Synergy of Intelligent Algorithms for Efficient Child-Robot Interaction in Special Education: A Feasibility Study,"International Conference on Robotics in Education (RiE), pp. 98-105, 2020.https://doi.org/10.1007/978-3-030-67411-3_9

We have added information regarding the processing speed of the employed frameworks at the end of the results section. More precisely, we added information regarding the used hardware, the time it takes for the proposed framework to process a single situation and all employed situations, and the time it takes for the Gibbs Sampling algorithm of the baseline to perform one iterations as well as all 100 iterations required to conclude the training.

Reviewer 3 Report

In the introductory chapter, the authors analyzed in detail the work dealing with the problem of robot-human interaction at the social level. This area currently requires more detailed attention to specific problems, as this behavioral interaction requires considerable mastery of problem-solving approaches such as cross-situational learning (CSL). It was the appropriate use of CSL modifications that made it possible to approach the possibilities of using artificial agents for the needs of appropriate learning of specific words.

In conclusion part authors state that manuscript investigated whether the proposed unsupervised online grounding framework is able to ground abstract concepts, like emotion types, emotion intensities and genders, during simulated human-agent interactions. Based on the examination of the results, it can be stated that the authors met the goal and presented in the required form in Chapter 7, the results of simulations that support the claims.

I consider the results obtained within the experimental part to be the strongest part of the article, which by their scope support the overall quality of the manuscript. All these results were realized on the basis of defined algorithms, within which the authors realized their modifications and adjustments.

Manuscript generally contains an interesting topic and an even more interesting solution.  I recommend publishing an article in Robotics journal.

Author Response

Dear reviewer,   thank you for your review.   Best regards, The authors

Round 2

Reviewer 2 Report

The authors tried to address the reviewer's concerns, but one of them needs additional clarification. More precisely:

The provided complexity analysis is of low importance since:

1) The training time is reported, while the inference time of the models is more important since it controls the real-time performance of the proposed methodology considering that the training is usually executed offline.

2) The provided performance is measured for the case of a computer running all the proposed models. 
However, when this methodology comes to be deployed as a part of an interaction between a social robot and a human, 
the performance is decreased as the two provided references examined under different configurations. 
Therefore, the question about the deployment issues of the proposed methodology is still unanswered.

Author Response

Dear reviewer,

thank you for your additional comments.

Please find our replies below. We have also made a few changes in the manuscript, which are again highlighted in green.

Best regards, The authors   1) The training time is reported, while the inference time of the models is more important since it controls the real-time performance of the proposed methodology considering that the training is usually executed offline

The inference time is for many machine learning models negligible because it is often in the realm of micro or milliseconds, while the training time is very important when considering deployment in dynamically changing environments, which are the environments social robots need to interact in, because it determines how easily new data can be integrated into the existing model. Furthermore, while it is true that many algorithms require an offline training phase, the proposed framework does not require any offline training but is able to update its mappings during the interaction so that the time to incorporate a new situation is very important to evaluate its deployability. Nevertheless, we added the inference time for both investigated models for interested future readers because it highlights why the proposed model is more applicable for deployment in dynamic environments by showing that it requires less than half the time to incorporate a new situation and infer the correct mappings than the baseline model requires for just inference.

2) The provided performance is measured for the case of a computer running all the proposed models. However, when this methodology comes to be deployed as a part of an interaction between a social robot and a human, the performance is decreased as the two provided references examined under different configurations.

Since the investigated frameworks can both be directly deployed on the used social robot, e.g. both Nao [1] and Pepper [2] from SoftBank Robotics provide with an Atom E3845 CPU @ 1.91 GHz x 4 and 4 GB RAM more than enough computational resources, there is no need to transmit any data to another computer for processing. Therefore, an analysis of the performance decrease due to network latency does not make much sense, especially since the introduced latency would also depend substantially on the employed data transfer mechanisms as well as the network setup, e.g. direct wireless connection or via a router.

[1] https://developer.softbankrobotics.com/nao6/nao-documentation/nao-developer-guide/technical-overview/motherboard

Round 3

Reviewer 2 Report

The authors have addressed all the reviewer's comments.